# DAPE: Dual-Stage Parameter-Efficient Fine-Tuning for Consistent Video Editing with Diffusion Models

## Abstract

Video generation based on diffusion models presents a challenging multimodal task, with video editing emerging as a pivotal direction in this field. Recent video editing approaches primarily fall into two categories: training-required and training-free methods. While training-based methods incur high computational costs, training-free alternatives yield suboptimal performance. To address these limitations, we propose **DAPE**, a high-quality yet cost-effective two-stage parameter-efficient fine-tuning (PEFT) framework for video editing. In the first stage, we design an efficient norm-tuning method to enhance temporal consistency in generated videos. The second stage introduces a vision-friendly adapter to improve visual quality. Additionally, we identify critical shortcomings in existing benchmarks including limited category diversity, imbalanced object distribution, and inconsistent frame counts. To mitigate these issues, we curate a high-quality dataset benchmark comprising more videos with rich annotations and editing prompts, enabling objective and comprehensive evaluation of advanced methods. Extensive experiments on existing datasets (BalanceCC, loveu-tgve, RAVE) and our proposed benchmark demonstrate that DAPE significantly improves temporal coherence and text-video alignment while outperforming previous state-of-the-art approaches. We will release the code and dataset in the final version.

## 1 Introduction

Video generation (Vondrick et al., 2016; Ho et al., 2022b; Singer et al., 2022; Ho et al., 2022a; OpenAI, 2024) has emerged as one of the most challenging and promising research directions within computer vision in recent years. As a prominent subfield, video editing (Wu et al., 2023a; Qi et al., 2023; Liu et al., 2024; Yang et al., 2025) aims to controllably modify the visual elements (e.g., objects, backgrounds), semantic information (e.g., textual descriptions), or dynamic characteristics (e.g., motion trajectories) of existing video contents while maintaining spatio-temporal coherence (Sun et al., 2024a). This technique holds significant commercial value, especially in areas such as metaverse and digital human creation, drawing considerable attention from leading technology companies like Microsoft (Feng et al., 2024; Xing et al., 2024), Google (Ho et al., 2022a), Nvidia (Blattmann et al., 2023b) and OpenAI (OpenAI, 2024). Figure 1 shows four applications.

Inspired by the recent success of diffusion models (Ho et al., 2020; Dhariwal & Nichol, 2021) and image editing methods (Hertz et al., 2022; Couairon et al., 2022; Brooks et al., 2023), contemporary video editing approaches typically adopt DDIM Inversion (Song et al., 2020) strategy and subsequently apply various conditioning strategies during denoising to facilitate content editing. For instance, RAVE (Kara et al., 2024) enhances temporal consistency via grid concatenation and noise shuffling for conditional injection, while CCEdit (Feng et al., 2024) improves the precise and creative editing capabilities by introducing a novel trident network structure that separates structure and appearance control. However, training-based methods generally incur high computational costs, whereas training-free methods typically struggle to achieve high-quality results. Balancing computational efficiency and video generation quality remains a critical challenge in video editing research (Sun et al., 2024b).

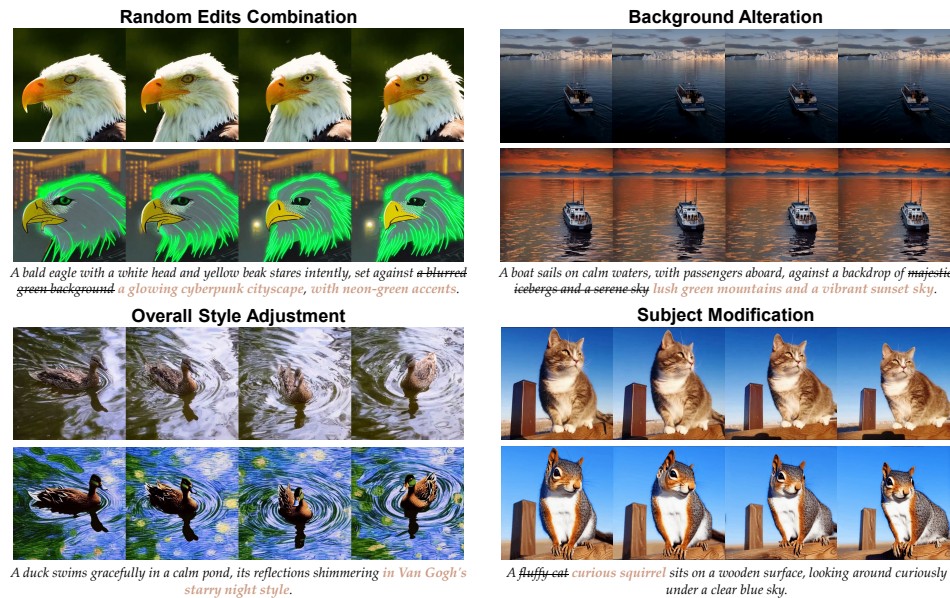

Figure 1: **DAPE** is a high-quality and cost-effective dual-stage parameter-efficient fine-tuning framework for text-based video editing. The diagram presents the performance of our method (lower) on original videos (upper) across four typical scenarios.

In visual tasks (Yin et al., 2023; 2024a;b), parameter-efficient fine-tuning (PEFT) techniques have been widely employed to enhance the performance of large-scale models on specific downstream tasks, such as image recognition (Zhang et al., 2020) and object segmentation (Peng et al., 2024). PEFT methods optimize only a small subset of model parameters, thus significantly reducing training costs and enhancing model performance even with limited training data (Houlsby et al., 2019; Xin et al., 2024). Diffusion-based video editing task often uses a single video template to generate new videos (Wu et al., 2023a; Kara et al., 2024), inherently forming a few-shot learning scenario (Song et al., 2023). Hence, leveraging PEFT to balance computational cost and video editing quality is highly promising. Despite its potential, PEFT remains under-explored in video editing, and it is essential to conduct a comprehensive investigation into its value within video editing tasks.

To address the challenges of optimizing video editing performance and computational efficiency, we propose DAPE, a novel dual-stage parameter-efficient fine-tuning approach for video editing designed to enhance temporal and visual consistency. First, recent studies have demonstrated that parameter plays a crucial role in enhancing textual condition controlling (Peebles & Xie, 2023; Huang & Belongie, 2017) and visual understanding (Basu et al., 2024), with recent evidence shows that temporal consistency in text-to-video (T2V) generation is particularly sensitive to normalization scales within temporal layers (Zhang et al., 2023b). To address this sensitivity, we propose a novel norm-tuning strategy and introduce a learnable scale factor to balance the original and normalized features optimally. Second, we find that adapter-based fine-tuning methods can effectively enhance the image quality of video editing. To improve model comprehension of single-video templates, we design a visual adapter module strategically integrated into the diffusion model. After that, we try to combine the strengths of both. In exploring the individual effects of these two optimization schemes, we find that separately, each significantly enhances either temporal consistency or visual quality. However, jointly training them introduces negative interactions, compromising their respective strengths. Considering our low-cost training methods with only single video, we employ a dual-stage framework to concurrently enhance both temporal consistency and visual quality. Furthermore, existing video editing benchmarks suffer from excessive frame lengths, low visual quality, and limited content diversity, thus inadequately assessing overall model capabilities. To address these limitations, we present a novel and comprehensive, DAPE Dataset, characterized by standardized format, high-quality visuals, and a wide variety of video types. The DAPE Dataset comprises 232 videos (significantly more than other video editing benchmarks commonly used in recent academic literature), each accompanied by a detailed video caption, video element types annotations,

video scene complexity labels, and a set of diverse editing prompts. Extensive experimental evaluations conducted on our DAPE Dataset and three representative benchmarks (RAVE Dataset (Kara et al., 2024), BalanceCC (Feng et al., 2024), loveu-tgve (Wu et al., 2023b)) demonstrate that our proposed method quantitatively and qualitatively outperforms previous state-of-the-art, substantially advancing temporal and visual consistency in video editing.

The key contributions are summarized as follows:

- We propose a novel dual-stage parameter-efficient fine-tuning method to significantly improve temporal and spatial consistency in video editing tasks.

- We design effective PEFT modules for the video editing tasks during each stage respectively, aiming to optimize temporal consistency and visual feature comprehension.

- We introduce a high-quality and comprehensive DAPE Dataset, enabling comprehensive and objective assessment of video editing methods.

- Extensive experiments on multiple datasets (DAPE Dataset, RAVE Dataset, BalanceCC, loveu-tgve) validate the superior performance of our method, outperforming previous state-of-the-art quantitatively and qualitatively.

## 2 RELATED WORK

**Text-Guided Video Editing** Text-guided video editing offers an efficient and lightweight alternative for video generation by adapting T2I diffusion models to modify video content while preserving original motion dynamics.This paradigm can be broadly categorized into two approaches, training-based and training-free. Training-based approaches typically fine-tune temporal layers of diffusion models to capture inter-frame temporal relationships, including Tune-A-Video (Wu et al., 2023a), Edit-A-Video (Shin et al., 2024), Video-P2P (Liu et al., 2024) and EI2 (Zhang et al., 2023b), while training-free methods often utilize frame-level feature guidance or auxiliary conditions (e.g., depth maps, sketches) to enhance consistency (such as Tokenflow (Geyer et al., 2023), FateZero (Qi et al., 2023), Render-A-Video (Yang et al., 2023), ControlVideo (Zhang et al., 2023a) and RAVE (Kara et al., 2024)). Here, we propose a fine-tuning-based video editing framework to optimize the temporal consistency and visual quality of existing methods.

**Parameter-Efficient Fine-Tuning** In natural language processing (NLP), PEFT techniques alleviate the computational overhead associated with fully fine-tuning models for downstream tasks by reducing the number of trainable parameters while maintaining performance. Recent investigations in video generation have also explored PEFT approaches. For instance, SimDA (Xing et al., 2024) efficiently adapted a 1.1B text-to-image model for video synthesis using only 24 million trainable parameters. ExVideo (Duan et al., 2024) achieved long-video generation by leveraging 3D convolutions and parameter-efficient post-tuning. T2V Adapter (He et al., 2025) proposed prompt-learning adapter GE-Adapter for video editing effect optimization. However, these PEFT methods are not specifically optimized for temporal consistency and image quality in video editing tasks.

## 3 METHODOLOGY

### 3.1 PRELIMINARIES

**Latent Diffusion Models (LDMs).** LDMs (Rombach et al., 2022) are efficient variants of DDPMs (Ho et al., 2020) that operate the diffusion process in a latent space. They are mainly built upon two key components. First, an auto-encoder maps images $x$ to the latent space $z = \mathcal{E}(x)$ and reconstructs them via $\mathcal{D}(z)$ enabling $\mathcal{D}(\mathcal{E}(x)) \approx x$. The diffusion process is then performed on $z$, using a U-Net based network to predict the added noise $\epsilon_\theta$. The objective of LDMs is as follows: where $z_t$ denotes the noisy latent at timestep $t$, and $c$ represents the text condition embedding.

**Adapter Tuning.** As a typical parameter-efficient fine-tuning method, adapter tuning refers to the approach that integrating small, trainable modules into models and fine-tuning them during training (Houlsby et al., 2019). These learnable structure can facilitate robust performance in specific downstream tasks by capturing domain-specific variations while avoiding catastrophic forgetting. A

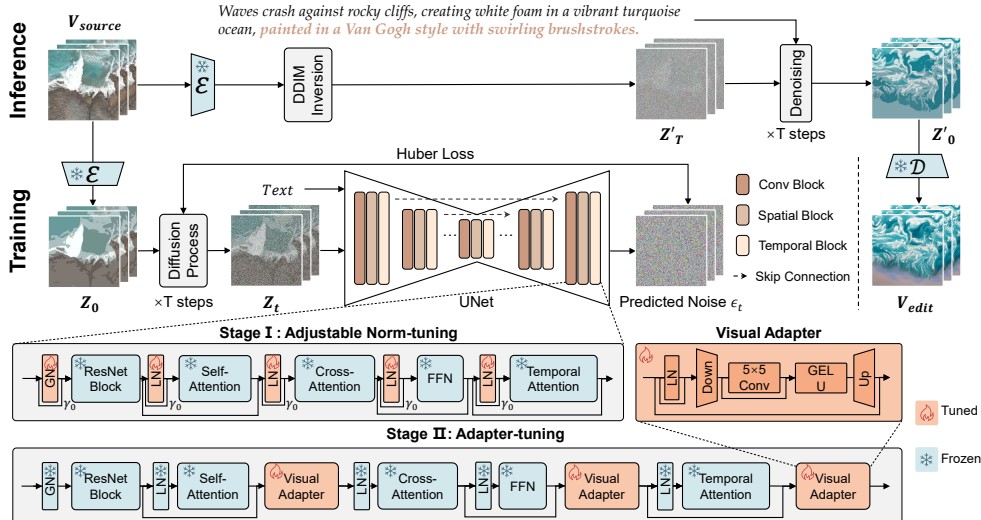

Figure 2: *Overall of DAPE.* DAPE is a generative model fine-tuning approach specifically designed for video editing tasks. In the first stage, we propose a novel norm-based fine-tuning method to enhance temporal consistency in editing results. In the second stage, we improve the visual quality of editing results through a carefully designed adapter fine-tuning approach..

conventional adapter module can be formulated as follows:

$$\text{Adapter}(\mathbf{X}) = \mathbf{X} + \mathbf{W}_{\text{up}}\left(\phi\left(\mathbf{W}_{\text{down}}(\mathbf{X})\right)\right), \qquad (1)$$

where $\mathbf{W}_{\text{down}}$ and $\mathbf{W}_{\text{up}}$ are the learnable projection matrices, and $\phi(\cdot)$ denotes an activation function.

### 3.2 FRAMEWORK

**DAPE Architecture.** As illustrated in Figure 2, DAPE is a diffusion-based framework that imporves both cross-frame temporal consistency and local-frame visual fidelity for video editing tasks. We adopts a dual-stage paradigm that decouples the learning of temporal and visual features to mitigating strength conflict. During training stage, we first exclusively fine-tunes the normalization layer parameters using our adjustable norm-tuning methods to establish a robust, temporally coherent backbone. Then we freezes the now-optimized normalization layers, insert and exclusively train the lightweight visual adapter to enhance visual quality and fine-grained details. During inference, we uses DDIM Inversion (Song et al., 2020) to retain the original video's features within the initial noise and progressively removes the U-Net-predicted noise conditioned on various inputs, ultimately generating the edited videos.

**Adjustable Norm-tuning.** Motivated by recent findings highlighting the pivotal role of normalization layers in shaping the quality and consistency of generation (Peebles & Xie, 2023; Zhang et al., 2023b), we introduce a novel approach, namely adjustable norm-tuning, to optimize normalization parameters of diffusion models blocks including ResNet blocks and attention blocks. To make the fine-tuning process dataset-specific, enhance feature scaling and enable better adaptation to new domains, a learnable affine parameters $\gamma_0$ is incorporated in the norm-tuning step. $\gamma_0$ is initialized to 0 as conventional normalization conduct and is multiplied on latent representations $z_t$. In the lower part of Figure 2, stage I can be formulated as follows:

$$\hat{z}_t = \gamma \cdot Norm(z_t) + \beta + \gamma_0 \cdot z_t, \qquad (2)$$

where $z_t$ is the input latent feature at timestep $t$, $Norm(\cdot)$ denotes a normalization operation with learnable parameters $\gamma, \beta$.

**Visual Adapter.** Adapters have been widely used to capture visual features in image tasks (Yin et al., 2023). To improve the stability of training and model adaptability, a layer normalization block with a learnable scaling parameters $w_0$ is adopted, followed by down projection, convolution layer,

nonlinear activation, up projection, and skip connections. Notably, to enhance spatial perceptual capabilities while minimizing additional parameters, convolution layer using a single depth-wise $5 \times 5$ kernel, leading to measurable improvements in extensive experiments. Besides, the visual adapter is placed exclusively within the first cross-attention block of the up-sampling (decoder) layers to achieve the best performance both in temporal coherence and alignment, shown in our ablation study. The procedure can be formally described as follows, also shown in Figure 2:

$$z = z_0 + \mathrm{Up}(\sigma(z_{\mathrm{conv}})),$$
$$z_{\mathrm{conv}} = \mathrm{Down}(z_{\mathrm{norm}}) + \omega_{dw} \otimes_{dw} \mathrm{Down}(z_{\mathrm{norm}}) \tag{3}$$

where $\sigma$ is the activation function, $\omega_{dw}$ denotes the convolutional kernel and $\otimes_{dw}$ indicates depth-wise convolution.

**Loss Function.** Considering that one-shot fine-tuning on a single video can create a significant domain discrepancy relative to the model's large-scale pre-training data, robustness to feature outliers is crucial. Therefore, we adopt the Huber loss, which combines the stability of MSE for small errors with the robustness of L1 loss for large errors, leading to more stable gradients. Our ablations confirm that Huber loss yields marginal but consistent gains over MSE for our task. The Huber loss is defined as:

$$\mathcal{L}_\delta(r) = \begin{cases} \frac{1}{2}r^2 & \text{if } |r| \le \delta, \\ \delta \cdot (|r| - \frac{1}{2}\delta) & \text{otherwise,} \end{cases} \tag{4}$$

where $r$ is the residual between the predicted and target noise, and $\delta$ is a threshold hyperparameter.

# 4 DAPE BENCHMARK

**Establishment.** Despite the availability of several datasets in the field of video editing, current benchmarks still suffer from key limitations including inconsistent resolution and frame count, low visual quality such as excessive camera motion and image blur, and limited content diversity. These flaws hinder researchers from conducting objective assessments of model performance. To mitigate evaluation bias, we introduce the **DAPE Dataset**, a standardized benchmark specifically designed to support video editing researches, which is significantly more than other benchmarks commonly used in recent academic literature.

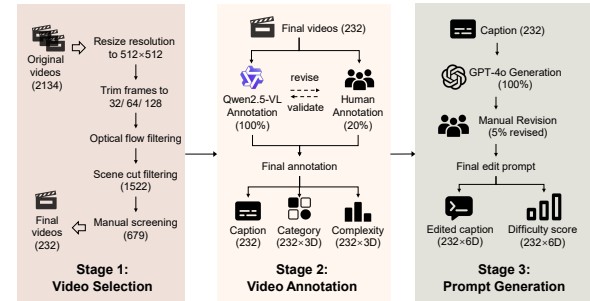

Figure 3: *The curation pipeline of our dataset.* We curated 232 high-quality samples by video selection, video annotation and human-in-the-loop annotations.

**Curation.** The curation pipeline of our dataset is illustrated in Figure 3. We curated 232 high-quality videos from an initial pool of over 2,100 commercially licensed sources (Kara et al., 2024; Xu et al., 2016; Wang et al., 2019; Pixabay). Each video underwent a rigorous filtering pipeline: it was first standardized to a 512x512 resolution and a duration of 32, 64, or 128 frames; then automatically filtered to remove excessive motion and scene discontinuities (Blattmann et al., 2023a; Wang et al., 2024); and finally manually verified for visual clarity and quality. These strict selection criteria and human-involved verification ensured that each video was suitable for effective evaluation. Moreover, the distribution of videos resulted from random sampling of a large pool of real-world videos, reflecting the typical characteristics of user-generated content. For annotation, we employed a human-in-the-loop approach: initial captions and diverse editing prompts were generated using state-of-the-art models (Qwen2.5, GPT-4o) (Teodoro et al., 2024; Feng et al., 2024; Bai et al., 2025; Hurst et al., 2024), and were subsequently reviewed by multiple human annotators to ensure accuracy and relevance.

**Statistics** The overall distribution of semantic categories and complexity levels in the DAPE Dataset is illustrated in Figure 4. The DAPE Dataset covers diverse semantic categories: "people" (33%) and "animal" (18%) are most common, while "artifact," "environment," "vehicle," and "food" constitute the remainder, reflecting the dominance of human-centric content. For background and events, distributions are more balanced, with indoor scenes (34%) and daily events (44%) most

frequent, consistent with everyday user-generated videos. Each component is further assigned a three-level complexity score (simple, moderate, complex), with the dataset emphasizing the first two to match current model capabilities. Supplementary material provides details on construction, annotation, sample visualizations, and dataset comparisons.

Figure 4: *Dataset statistics.* Distributions of the DAPE Dataset across six semantic dimensions: category and complexity for subject, background, and event.

## 5 EXPERIMENTS

### 5.1 SETTINGS

**Implementation Details.** DAPE employs the pre-trained T2I model, stable diffusion-v1.5 as initialization weights. Adjustable norm-tuning stage employs 400 timesteps with a learning rate of $5 \times 10^{-5}$ and a batch size of 1, while the visual adapter tuning stage involves 70 timesteps at a learning rate of $1 \times 10^{-5}$ with the same batch size. During inference, the sampler configured for 50 steps, classifier-free guidance (Ho & Salimans, 2022) 7.5. Our experiments are conducted on 8 NVIDIA A800 GPUs.

**Baselines.** We select five latest baseline methods covering both training-based and training-free approaches using their official implementations, including **Tune-A-Video (ICCV'23)** (Wu et al., 2023a), **CAMEL (CVPR'24)** (Zhang et al., 2024), **SimDA (CVPR'24)** (Xing et al., 2024), **RAVE (CVPR'24)** (Kara et al., 2024), and **CCEdit (CVPR'24)** (Feng et al., 2024). Our proposed DAPE framework can also be applied to other frameworks. Therefore, we conducted many experiments based on each baseline to demonstrate the potential insights and implications of our approach for other models.

**Datasets.** To fully demonstrate the effectiveness of our methods, we conduct experiments on our **DAPE Dataset** and three other lastest and typical video editing datasets: loveu-tgve (Wu et al., 2023b), RAVE Dataset (Kara et al., 2024), and BalanceCC (Feng et al., 2024).

**Evaluation Metrics.** We evaluate generated videos primarily from two perspectives: 1) **Temporal consistency**: it consists of CLIP-Frame calculating the average CLIP score pairwise similarity across frames, Interpolation Error and PSNR (Jiang et al., 2018) and Warping Error (Lai et al., 2018) employing RAFT (Teed & Deng, 2020). 2) **Text-video alignment**: we use the widely adopted metric CLIP-Text to assess text-video alignment, computing the mean similarity between video frame embeddings and textual embeddings via the CLIP model (Radford et al., 2021).

### 5.2 MAIN RESULTS

**Quantitative Results.** Table 1 presents the quantitative results of all methods on the four datasets. Macroscopically, DAPE achieves the best performance (highlighted in bold) across all datasets, demonstrating its effectiveness in enhancing the quantitative performance of mainstream video editing tasks. Microscopically, DAPE significantly improves the performance of baseline methods on most metrics. These results confirm that the proposed adjustable norm tuning and visual adapter components , as integral elements of our framework, effectively enhance temporal consistency and alignment. Additionally, our analysis of the results on the three existing datasets reveals that the ranking of baseline methods varies across different datasets. This observation further underscores the necessity of establishing a comprehensive benchmark dataset.

**Qualitative Results.** Figure 7 illustrates the marked differences among the methods regarding temporal consistency, text alignment, and detail quality. For instance, when tasked with changing an SUV to a red sports car, baselines either fail on color/object accuracy (TAV, CAMEL, CCEdit) or quality (SimDA, RAVE), whereas DAPE successfully generates a high-fidelity result. Similarly, in applying a Van Gogh style, most methods fail to synthesize the artistic effect, while DAPE produces a consistent and appealing stylized video. For object replacement (squirrel to rabbit), competing methods introduce artifacts like inconsistent identity (TAV), poor details (CAMEL, RAVE), or se-

| Method | BalanceCC | | | | | loveu-tgve | | | | |
|---|---|---|---|---|---|---|---|---|---|---|
| | Temporal Consistency | | | | Alignment | Temporal Consistency | | | | Alignment |
| | C. F. ↑ $\times 10^{-2}$ | I. E. ↓ $\times 10^{-2}$ | I. P. ↑ $\times 1$ | W. E. ↓ $\times 10^{-2}$ | C. T. ↑ $\times 10^{-2}$ | C. F. ↑ $\times 10^{-2}$ | I. E. ↓ $\times 10^{-2}$ | I. P. ↑ $\times 1$ | W. E. ↓ $\times 10^{-2}$ | C. T. ↑ $\times 10^{-2}$ |
| Baselines | | | | | | | | | | |
| TAV [ICCV'23] | 93.11 | 14.43 | 17.57 | 5.57 | 31.82 | 94.44 | 10.36 | 20.82 | 4.05 | 29.95 |
| CAMEL [CVPR'24] | 94.67 | 8.80 | 22.61 | 4.07 | 29.27 | 94.44 | 10.14 | 21.11 | 4.03 | 27.70 |
| SimDA [CVPR'24] | 91.32 | 12.79 | 18.57 | 5.06 | 31.28 | 91.96 | 9.08 | 21.75 | 3.11 | 29.33 |
| RAVE [CVPR'24] | 94.10 | 8.69 | 22.05 | _2.46_ | _32.15_ | 94.32 | 8.27 | 22.59 | _2.34_ | _30.18_ |
| CCEdit [CVPR'24] | _95.50_ | _7.29_ | _24.33_ | 4.52 | 29.76 | 94.00 | _7.65_ | _23.76_ | 3.24 | 28.80 |
| Ours | | | | | | | | | | |
| DAPE (TAV) | 93.46 | 13.98 | 17.83 | 5.31 | 31.82 | _94.53_ | 10.28 | 20.88 | 3.96 | 29.98 |
| DAPE (CAMEL) | 94.75 | 8.68 | 22.75 | 3.94 | 29.37 | **94.67** | 10.04 | 21.20 | 4.07 | 27.78 |
| DAPE (SimDA) | 91.43 | 12.29 | 18.85 | 4.92 | 31.37 | 92.07 | 8.91 | 21.96 | 3.01 | 29.17 |
| DAPE (RAVE) | 94.61 | **7.18** | 23.91 | **2.13** | **32.85** | 94.33 | 7.73 | 23.16 | **2.18** | **30.35** |
| DAPE (CCEdit) | **95.54** | 7.58 | **24.38** | 4.03 | 30.19 | 93.76 | **7.59** | **23.85** | 2.97 | 29.32 |

| Method | RAVE Dataset | | | | | DAPE Dataset | | | | |
|---|---|---|---|---|---|---|---|---|---|---|
| | Temporal Consistency | | | | Alignment | Temporal Consistency | | | | Alignment |
| | C. F. ↑ $\times 10^{-2}$ | I. E. ↓ $\times 10^{-2}$ | I. P. ↑ $\times 1$ | W. E. ↓ $\times 10^{-2}$ | C. T. ↑ $\times 10^{-2}$ | C. F. ↑ $\times 10^{-2}$ | I. E. ↓ $\times 10^{-2}$ | I. P. ↑ $\times 1$ | W. E. ↓ $\times 10^{-2}$ | C. T. ↑ $\times 10^{-2}$ |
| Baselines | | | | | | | | | | |
| TAV [ICCV'24] | 94.35 | 15.03 | 16.64 | 5.55 | _31.09_ | 94.88 | 9.00 | 21.73 | 2.73 | 31.34 |
| CAMEL [CVPR'24] | 92.85 | 14.18 | 17.36 | 5.66 | 27.40 | 95.74 | 6.78 | 24.53 | 2.28 | 29.95 |
| SimDA [CVPR'24] | 91.94 | 13.75 | 17.43 | 5.40 | 30.07 | 92.22 | 7.96 | 22.75 | 2.42 | 30.61 |
| RAVE [CVPR'24] | _94.85_ | 8.71 | 21.94 | _2.53_ | 29.76 | 95.80 | 6.65 | 24.09 | _1.37_ | _32.52_ |
| CCEdit [CVPR'24] | 93.74 | 10.34 | 20.37 | 4.46 | 26.41 | _96.47_ | _5.41_ | _26.66_ | 1.90 | 28.56 |
| Ours | | | | | | | | | | |
| DAPE (TAV) | 94.53 | 14.77 | 16.78 | 5.40 | **31.19** | 94.92 | 9.14 | 21.80 | 2.66 | 31.47 |
| DAPE (CAMEL) | 92.93 | 14.10 | 17.43 | 5.47 | 27.41 | 95.89 | 6.61 | 24.74 | 2.21 | 30.00 |
| DAPE (SimDA) | 92.05 | 13.62 | 17.57 | 5.26 | 30.13 | 93.11 | 7.74 | 23.23 | 2.37 | 31.15 |
| DAPE (RAVE) | **94.98** | **8.34** | _22.30_ | 2.42 | 29.78 | 95.85 | 6.27 | 24.63 | **1.26** | **32.61** |
| DAPE (CCEdit) | 93.83 | _8.47_ | **22.37** | 2.90 | 28.35 | **96.59** | **5.31** | **27.09** | 1.52 | 29.07 |

Table 1: *Quantitative comparison.* Experimentsare conducted on four datasets to evaluate the models' performance on five metrics (CLIP-Frame (C. F.), Interpolation Error (I. E.), Interpolation PSNR (I. P.), WarpError (W. E.), CLIP-Text (C. T.)). ↑ means higher is better while ↓ donates the lower the better. The best/second-best performance are highlighted in **bold**/_underline_, respectively.

mantic errors (CCEdit). In contrast, DAPE maintains high-level consistency and fine-grained detail. In short, qualitative results indicate that our proposed DAPE method outperforms the baselines in terms of temporal consistency, text alignment and detail fidelity, ultimately leading to noticeably improved visual smoothness and semantic relevance.

**User Study.** We conduct a user study to further validate our method. A total of 1,536 responses were collected from 30 participants, each completing a questionnaire with 25 sets of comparisons. These participants have received a good aesthetic education, including individuals deeply specialized in computer vision and those with cross-disciplinary background. Participants are asked to rank the top-three videos by textual alignment, temporal smoothness, and visual quality. Our method outperforms the baselines, showing better alignment with human judgment. Figure 6 shows the results and the questionnaire example are provided in the appendix. The results of the user research indicate that the proposed method achieved superior performance across all three dimensions.

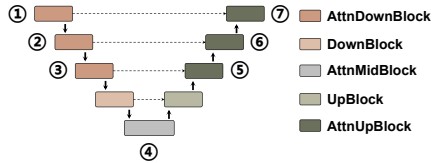

Figure 5: *Different blocks in U-Net.* For better clarity, we index U-Net blocks from ① to ⑦.

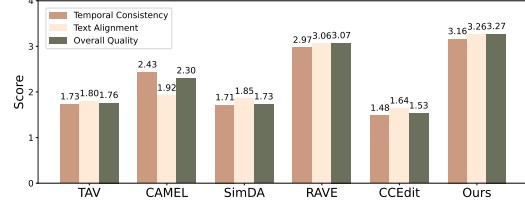

Figure 6: *User Study Results.* Comparison of subjective scores for each model. Models perform the best, second best and third best with 6, 5 and 4 scores, and the scores for each model are weighted by vote frequency.

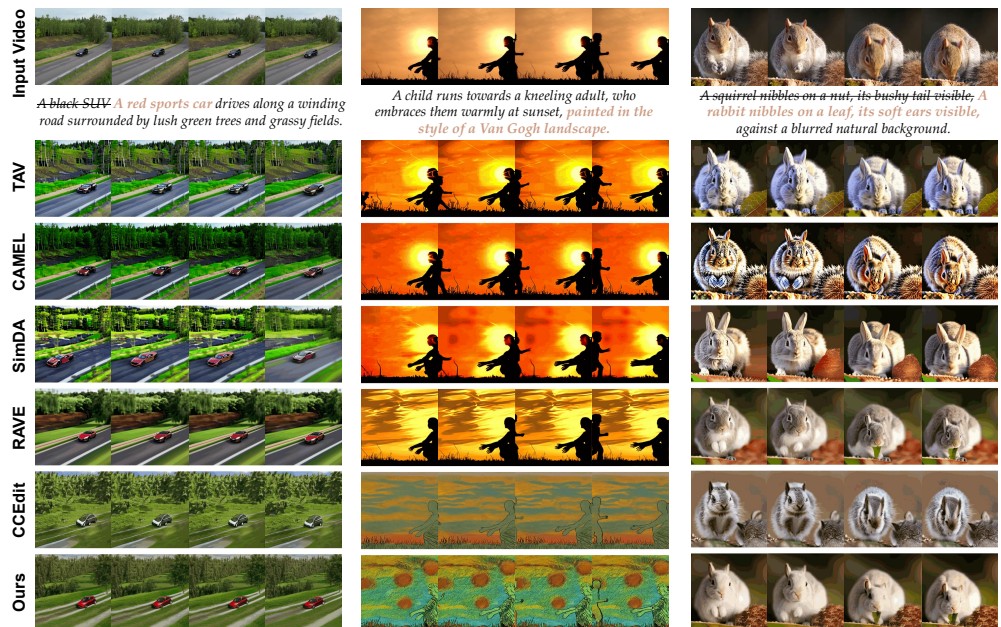

Figure 7: *Qualitative comparison.* Different model performance on typical editing tasks. DAPE performs better in terms of temporal consistency, text alignment and visual quality.

## 5.3 ABLATION STUDY

In this section, we conduct ablation experiments to validate our key design choices, including the adapter's embedding location, the impact of each module, and our framework's performance against other PEFT methods. All ablation experiments are conducted on DAPE dataset.

**Adapter Position.** We explored various adapter insertion points within the U-Net's attention blocks (see Figure 5) to identify the optimal configuration. As shown in Table 2, different placements yield a trade-off: inserting adapters in shallow layers (①②⑥⑦) boosts semantic consistency (CLIP-F) at the cost of structural coherence (Int. Err.), while using too many adapters (①-⑦) even degrades overall performance. Ultimately, placing the adapter solely in the first block of the decoder (⑤) achieves the best balance across all metrics. This configuration effectively refines high-level visual concepts without disrupting the features encoded in earlier layers. This suggests that the decoder's early layers are crucial for reconstructing high-level semantics from the latent bottleneck, making them the optimal point to influence core visual concepts (e.g., style, identity) with minimal disruption to the original content structure. Figure 8 shows an example among six settings.

**Module Impact and Design Choices.**
To validate our dual-stage design, we analyze the contribution of each component in Table 3. The Visual Adapter (V. A.) alone excels at improving visual details (lowest Int. Err.), while Adjustable Norm-tuning (A. N.) is more effective for temporal consistency and text alignment (highest CLIP-F and CLIP-T). Crucially, training both modules simultaneously in a one-stage setup results in suboptimal performance, confirming a negative interaction between the two optimization goals. Our proposed dual-stage approach successfully mitigates this conflict,

Table 2: *Ablation Results of Adapter Position.* ⑤ is selected as the final setting due to its balance of two judging dimensions.

| Method | Temporal Consistency | | | | Align. |
|---|---|---|---|---|---|
| | C. F. ↑ $\times 10^{-2}$ | I. E. ↓ $\times 10^{-2}$ | I. P. ↑ $\times 1$ | W. E. ↓ $\times 10^{-2}$ | C. T. ↑ $\times 10^{-2}$ |
| ①-⑦ | 94.59 | 9.10 | 21.57 | 2.71 | 28.66 |
| ①②⑥⑦ | **94.88** | 9.41 | 21.41 | 3.00 | 29.00 |
| ③-⑤ | 94.78 | 8.69 | 21.96 | 2.55 | 29.52 |
| ①-③ | 94.78 | 9.18 | 21.69 | 2.94 | 29.02 |
| ⑤-⑦ | 94.68 | 8.68 | 21.93 | 2.55 | 29.34 |
| ⑤ | 94.81 | **8.62** | **21.98** | **2.50** | **29.57** |

achieving the best overall performance, particularly in reducing Warping Error. Furthermore, we

Table 3: *Ablation on inner design of DAPE. The proposed two-stage setting can outperform baseline on all metrics. H. L., V. A., and A. N. stand for Huber Loss, Visual Adapter, and Adjustable Norm-tuning, respectively.*

| Method | Temporal Consistency | | | | Align. |
| | C. F. ↑ $\times 10^{-2}$ | I. E. ↓ $\times 10^{-2}$ | I. P. ↑ $\times 1$ | W. E. ↓ $\times 10^{-2}$ | C. T. ↑ $\times 10^{-2}$ |
|---|---|---|---|---|---|
| w/o All | 94.85 | 8.71 | 21.94 | 2.53 | 29.76 |
| w/o H. L. | 94.78 | 8.36 | 22.34 | 2.44 | 29.52 |
| w/ V. A. | 94.71 | **8.25** | **22.58** | 2.47 | 29.34 |
| w/ A. N. | **95.05** | 8.69 | 22.01 | 2.62 | **29.82** |
| One-stage | 94.76 | 8.37 | 22.51 | 2.50 | 29.42 |
| w/ all | 94.98 | 8.34 | 22.30 | **2.42** | 29.78 |

Table 4: *Quantitative comparison against PEFT baselines. DAPE outperforms all baselines across nearly every metric, including popular methods like LoRA and Adapter.*

| Method | Temporal Consistency | | | | Align. |
| | C. F. ↑ $\times 10^{-2}$ | I. E. ↓ $\times 10^{-2}$ | I. P. ↑ $\times 1$ | W. E. ↓ $\times 10^{-2}$ | C. T. ↑ $\times 10^{-2}$ |
|---|---|---|---|---|---|
| Fixed | 94.85 | 8.71 | 21.94 | 2.53 | 29.76 |
| LoRA | 94.54 | 10.55 | 20.45 | 3.58 | 29.32 |
| Adapter | 94.76 | 9.23 | 21.54 | 2.92 | 29.60 |
| Mona | 94.64 | 9.21 | 21.55 | 2.84 | 28.96 |
| Partial-1 | 94.81 | 8.78 | 21.84 | 2.56 | 29.71 |
| Norm-tuning | 94.92 | 8.53 | 22.24 | 2.57 | **29.92** |
| Bitfit | 94.85 | 9.00 | 21.71 | 2.75 | 29.78 |
| **DAPE** | **94.98** | **8.34** | **22.30** | **2.42** | 29.78 |

confirmed that using Huber Loss provides a consistent improvement over standard MSE loss (as shown in the last two rows of Table 3), making the one-shot tuning process more robust.

**Comparison with PEFT Baselines.** To situate DAPE within the broader PEFT landscape, we compare it against several standard fine-tuning methods in Table 4. The results clearly show that DAPE outperforms all baselines, including popular methods like LoRA and Adapter, across nearly every metric. Notably, while Norm-tuning alone is a strong performer, our full dual-stage framework further enhances the results, especially in temporal consistency (e.g., Warping Error) and visual quality (e.g., Int. PSNR).

## 6 CONCLUSION

In this paper, we introduce DAPE, a dual-stage parameter-efficient fine-tuning framework with adjustable norm-tuning and a carefully positioned visual adapter, to significantly enhance the temporal consistency and visual quality and generate more consistent videos. Accompanying this framework, we propose DAPE Dataset, a comprehensive benchmark designed to systematically evaluate performance across diverse editing scenarios. Extensive experimental validation confirmed that our approach achieves state-of-the-art results, effectively balancing visual quality, temporal coherence, and prompt adherence, paving the way for future research in generative model optimization and broader applications.

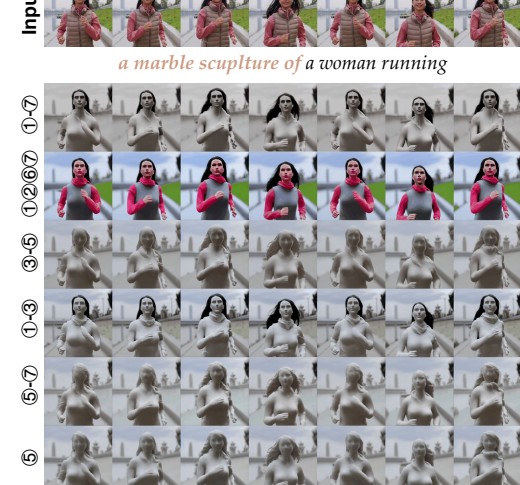

Figure 8: *Visualization of adapter ablation. The editing prompt requires changing the visual style to a marble sculpture. ①–⑦, ①②⑥⑦, and ①–③ fail to effectively follow the editing instruction. ③–⑤ negatively impact the facial lighting details, while ⑤–⑦ struggle to maintain temporal consistency. ⑤ achieves the optimal editing results.*

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

# A APPENDIX

## A.1 DATASET

**Dataset Statistics.** The overall distribution of semantic categories and complexity levels in the DAPE Dataset is illustrated in Figure 4. For subject type, the "people" category is the most prevalent (33%), followed by "animal" (18%), while "artifact," "environment," "vehicle," and "food" collectively make up the remainder. This designed choice reflects the dominance of human-centric content in real-world video scenarios. Regarding background and event types, the distribution is relatively balanced. Indoor scenes appear most frequently (34%), and "daily" events are the most common (44%), aligning with the characteristics of everyday user-generated content. Each of the three components is further annotated with a three-level complexity score: *simple, moderate, and complex*. The dataset is intentionally constructed to emphasize simple and moderate levels across all dimensions, considering the current maturity of video editing models.

**Dataset Construction.** The construction of our dataset is organized into three sequential steps: video selection, video annotation, and prompt generation, as illustrated in Figure 3. This pipeline is specifically designed for building video editing datasets, integrating both automated tools and human validation. Each video is accompanied by detailed annotations and multiple prompts for video editing tasks, as illustrated in Figure 9. The details of the three-step construction pipeline as follows.

**Step 1. Video Selection.** We initially collected 2,134 videos from four sources, including RAVE Kara et al. (2024), MSR-VTT Xu et al. (2016), VATEX Wang et al. (2019), and Pixabay. For large-scale datasets such as VATEX, random sampling was applied to reduce redundancy. Videos were resized to 512×512 and trimmed to 32, 64, or 128 frames. We then apply optical flow filtering to exclude samples with excessive motion and scene cut filtering to remove videos with abrupt transitions Blattmann et al. (2023a); Wang et al. (2024).

After automated filtering, 31.8% of the videos remained. These candidates then underwent manual quality screening, resulting in a curated set of 232 high-quality videos suitable for downstream evaluation. A video was retained only if it met the following predefined criteria across four key dimensions:

- **Motion**: Both camera and subject movement should be smooth and stable, without abrupt shaking or erratic motion.
- **Editing:** The video should maintain temporal continuity, with no scene cuts, montage transitions, or long static frames.
- **Content:** The visual subject must be complete and unobstructed, with no prominent overlaid text or distracting visual elements.
- **Visual Quality:** The overall presentation should be aesthetically coherent, with appropriate lighting, contrast, and composition.

| Dataset | #Videos | Frames | Resolution | #Prompts | Caption | #Sub Cats. | #Bg Cats. | #Evt Cats. |
|---|---|---|---|---|---|---|---|---|
| TGVE (CVPR'23) | 76 | 32/128 | 480×480 | 4 | Y | N/A | N/A | N/A |
| BalanceCC (CVPR'24) | 100 | 60–600 | 512×512 | 4 | Y | 4 | N/A | N/A |
| RAVE (CVPR'24) | 41 | 8/36/90 | mixed | 5 | N | 5 | N/A | N/A |
| MIVE (arXiv'24) | 200 | 12–46 | 512×512 | 2 | Y | N/A | N/A | N/A |
| **Ours** | **232** | **32/64/128** | **512×512** | **6** | **Y** | **6** | **4** | **5** |

Table 5: *Comparison of DAPE with existing video editing datasets.* 'Sub Cats.', 'Bg Cats.', and 'Evt Cats.' refer to subject categories, background categories and event categories, respectively.

**Step 2. Video Annotation.** We first conducted manual annotations on approximately 20% of the dataset, performed by three domain experts following a shared annotation guideline. These human-labeled samples served as a reference for calibrating the annotation quality. Based on this subset, we employed the Qwen2.5 vision-language model to annotate the remaining videos. For each video, eight evenly spaced frames were sampled as input to Qwen2.5, which was prompted to generate a caption, three semantic labels (subject, background, event), and corresponding complexity scores. Evaluation against the human-labeled subset showed that Qwen2.5 achieved an accuracy of approximately 97.9% in semantic classification, which was within acceptable bounds for large-scale annotation. Detailed examples of the videos and their corresponding annotations are illustrated in Figure 10.

Each video in our dataset is categorized based on three core components—subject, background, and event. The specific category sets for each component are adapted from the classification scheme used in MSR-VTT Xu et al. (2016), with modifications to better suit our video editing context.

- **Subject:** Indicates the primary entity or focus present in the video, including *people*, *animal*, *vehicle*, *artifact*, *food* and *environment*.
- **Background:** Describes the dominant scene or setting in which the video takes place, including *indoor*, *urban*, *natural* and *blur or blank*.
- **Event:** Refers to the main activity or situation depicted in the video, including *sports*, *daily*, *performance*, *documentary* and *cooking*.

**Step 3. Prompt Generation.** Compared with annotation, the task of generation is more challenging for human annotators. To address this, we leveraged the GPT-4o model to automatically generate editing instructions based on a designed framework. Each instruction targets one of five aspects: subject modification, background alteration, event reorganization, overall style adjustment, or a random combination thereof. The model produces an edited version of the caption and assigns a corresponding difficulty score for each prompt. All generated outputs were manually reviewed for feasibility. Domain experts further examined the instructions, 5.2% of prompts that were judged to be unrealistic were identified and revised.

Each video in our dataset is associated with five types of editing tasks, each targeting different aspects of the video content.

- **Subject Modification:** Alters the appearance or identity of the primary subject in the video such as *changing clothing*, *replacing a person with an animal*.
- **Background Alteration:** Modifies the visual setting or environment in which the video takes place such as *changing a kitchen scene to a grassland*.
- **Event Reorganization:** Modifies the main action or activity depicted in the video such as *changing a person walking a dog to playing basketball*.
- **Overall Style Adjustment:** Changes the visual tone or artistic style of the video such as *applying cartoon effects*, *converting to black-and-white*.
- **Random Edits Combination:** Randomly applies a combination of two editing types selected from the four categories above.

**Comparison with existing datasets.** In recent years, a surge of excellent work in video editing has also led to the release of several specialized datasets. For instance, the work on Tune-A-Video Wu et al. (2023a) explored one-shot tuning for text-to-video generation. Concurrently, methods like CCEdit Feng et al. (2024), which enables creative control by decoupling structure and appearance, RAVE Kara et al. (2024), which uses noise shuffling for fast and consistent editing,

and MIVE Teodoro et al. (2024), which tackles multi-instance editing, have all contributed to the landscape of available video editing benchmarks.

While these pioneering works and their associated datasets have driven progress, they exhibit certain limitations in scale, standardization, and annotation depth. As shown in Tab. 5, our DAPE dataset demonstrates clear advantages across multiple key metrics. Specifically, DAPE not only provides a larger collection of videos but also maintains a higher degree of standardization with consistent spatial (512x512) and temporal (32/64/128 frames) resolutions.

Crucially, DAPE surpasses existing datasets in annotation richness. All samples include natural language captions to support the effective training and evaluation of text-video alignment. Going a step further, DAPE introduces structured semantic annotations across subject, background, and event dimensions, encompassing 6, 4, and 5 categories, respectively. This fine-grained annotation scheme enables a more nuanced analysis of model capabilities. Furthermore, DAPE offers a broader range of editing prompt types, facilitating a more comprehensive evaluation of models under diverse and realistic editing scenarios.

## A.2 USER STUDY

To evaluate the performance of our proposed DAPE approach against existing methods (CCEdit, RAVE, SimDA, CAMEL, and TAV), we conducted a comprehensive user study. We recruited 30 anonymous participants, all of whom possess a strong background in aesthetic judgment, cultivated through formal education or professional experience. The participant pool was strategically composed of individuals with significant expertise in computer vision or related fields to ensure a high-quality and insightful evaluation. Specifically, our cohort included 3 professors/senior researchers, 13 Ph.D. students, and 14 Master's students, all with research backgrounds or practical experience in computer science. The study centered on 21 randomly selected video-text pairs from our dataset. Participants were asked to assess the editing results based on three key criteria: textual alignment, temporal consistency, and overall editing quality. As shown in 6, our method outperforms the baselines, illustrating better human intuition following, temporal continuity and visual fineness. An example of the questionnaire administered is shown in Figure 11.

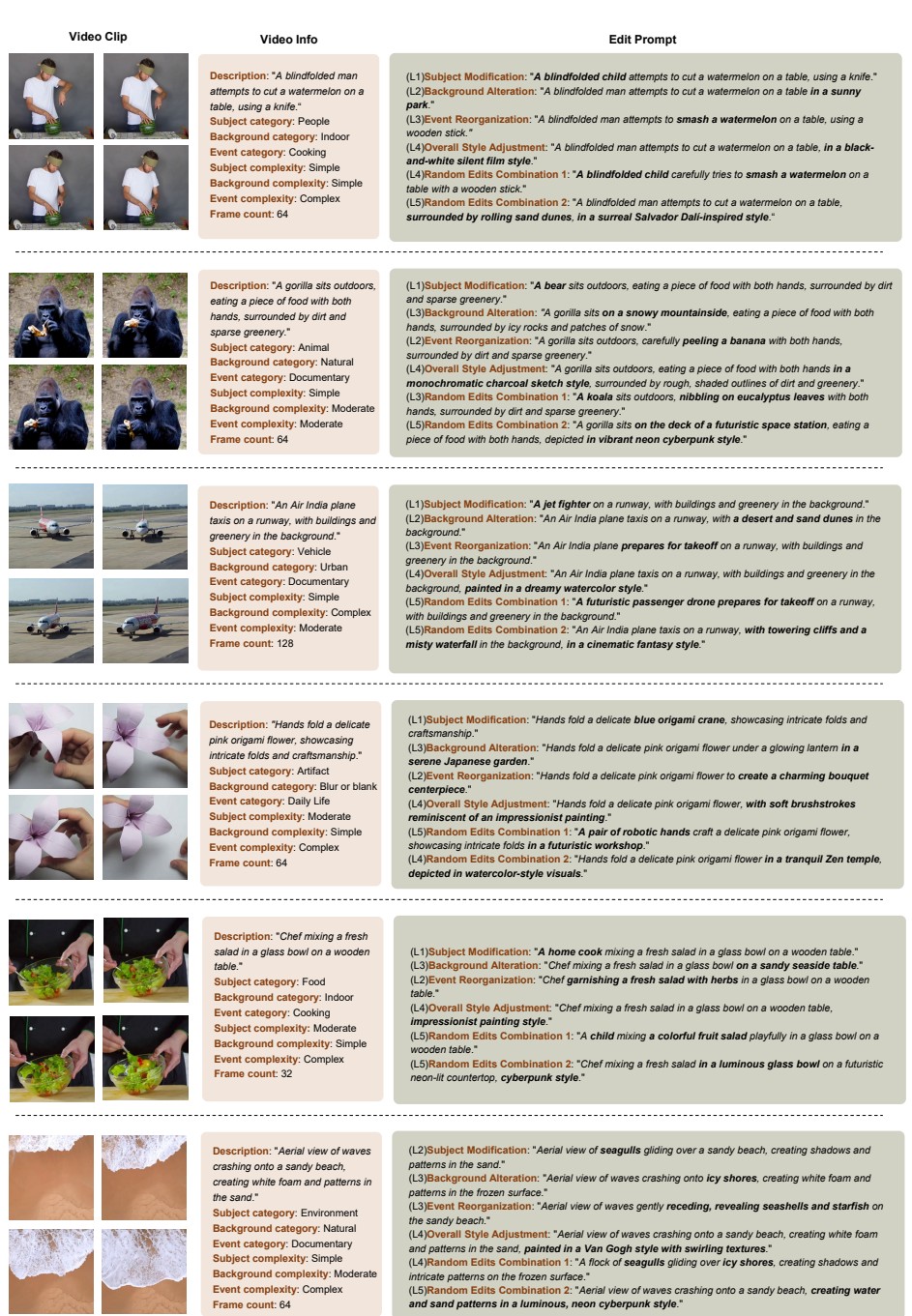

Figure 9: *Illustrative examples of our DAPE Dataset.* The labels (L1–L5) indicate the difficulty levels of the editing tasks.

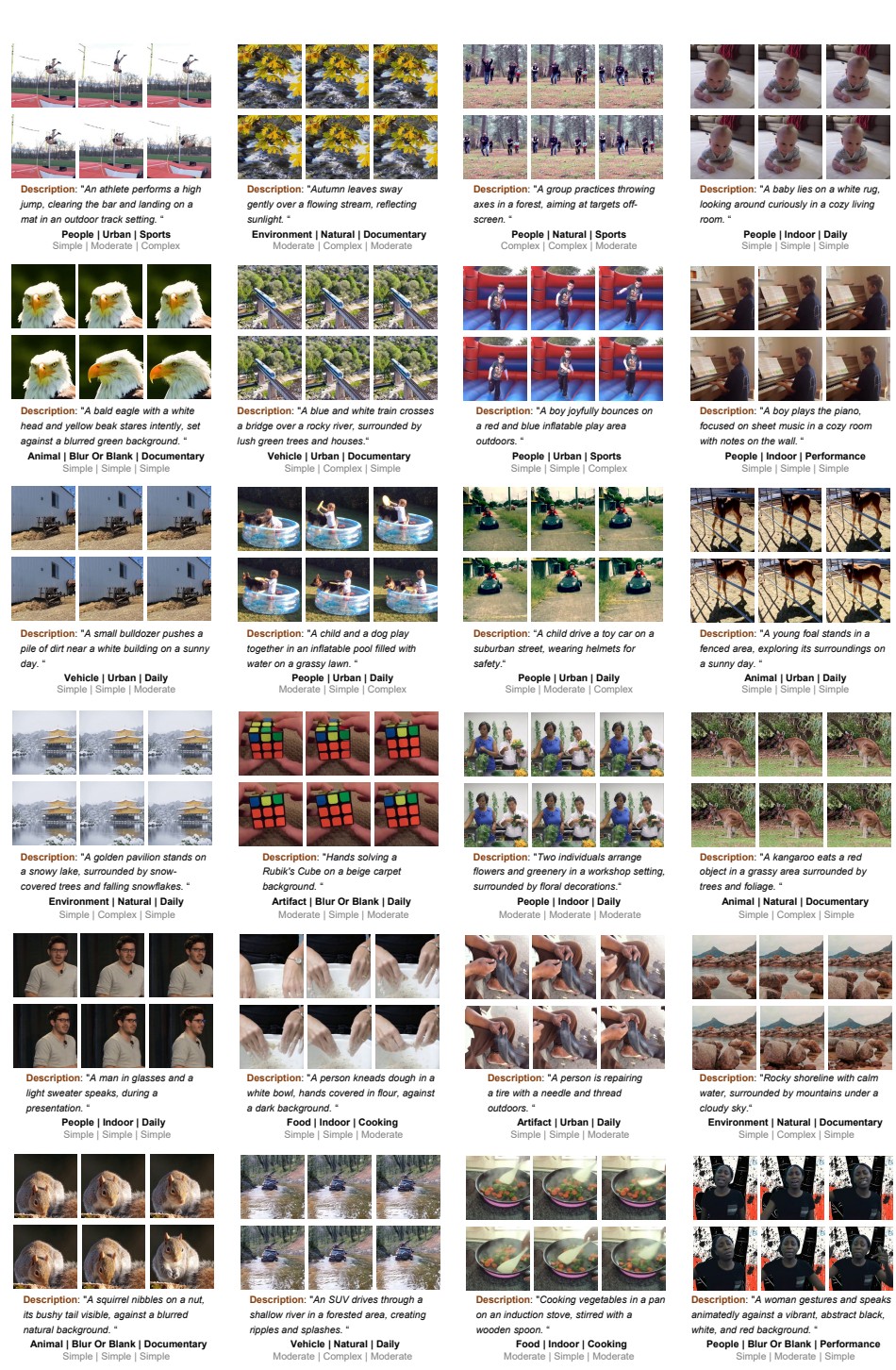

Figure 10: *More sample video frames from our DAPE Dataset.*

**We aim to evaluate the difference between the video generated by the model and the original video, so as to compare the advantages and disadvantages of the methods. Please answer the corresponding questions according to your visual perception. It will be evaluated from three aspects: instruction compliance, video fluency and overall effect, including 21 videos in total, which is expected to take 15-20min.**

**Input Video**

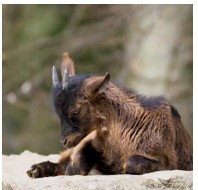

**Text Prompt: "A young goat sits on a rock, grooming itself with its front paw, in a hand-drawn watercolor painting style"**

| | 1 | 2 | 3 | 4 | 5 | 6 |

**1.Please select and rank the top three most satisfactory generated videos according to their compliance with the text instructions.**

| | Video 1 | Video 2 | Video 3 | Video 4 | Video 5 | Video 6 |
|--------|---------|---------|---------|---------|---------|---------|
| Best | ○ | ○ | ○ | ○ | ○ | ○ |
| Second | ○ | ○ | ○ | ○ | ○ | ○ |
| Third | ○ | ○ | ○ | ○ | ○ | ○ |

**2.Please choose according to the overall smoothness of the generated video (no distortion, flicker, etc.), select the top three most satisfactory videos and rank them.**

| | Video 1 | Video 2 | Video 3 | Video 4 | Video 5 | Video 6 |
|--------|---------|---------|---------|---------|---------|---------|
| Best | ○ | ○ | ○ | ○ | ○ | ○ |
| Second | ○ | ○ | ○ | ○ | ○ | ○ |
| Third | ○ | ○ | ○ | ○ | ○ | ○ |

**3.Please select and rank the top three most satisfactory generated videos according to the overall visual experience of the generated videos.**

| | Video 1 | Video 2 | Video 3 | Video 4 | Video 5 | Video 6 |
|--------|---------|---------|---------|---------|---------|---------|
| Best | ○ | ○ | ○ | ○ | ○ | ○ |
| Second | ○ | ○ | ○ | ○ | ○ | ○ |
| Third | ○ | ○ | ○ | ○ | ○ | ○ |

Figure 11: *Questionnaire example of user study.*

