# OpenReview forum: "DAPE: Dual-Stage Parameter-Efficient Fine-Tuning for Consistent Video Editing with Diffusion Models"
_ICLR.cc/2026/Conference — ICLR 2026 Conference Withdrawn Submission_

### Official Review · Reviewer_bs3G · 2025-10-28

**Soundness:** 3
**Presentation:** 4
**Contribution:** 3
**Rating:** 4
**Confidence:** 4

**Summary:**

This paper addresses the challenge of high computational costs and suboptimal performance in diffusion-model-based video editing. The authors propose DAPE, a dual-stage parameter-efficient fine-tuning (PEFT) framework designed to improve both video quality and temporal consistency cost-effectively.

The paper's main contributions are twofold: 1) The DAPE framework itself, which is shown to be a general method that can improve existing video editing models (e.g., RAVE, CCEdit). 2) The curation of a new, high-quality benchmark, the "DAPE Dataset," which contains 232 videos with rich annotations and diverse editing prompts, addressing limitations of existing benchmarks.

**Strengths:**

1. The paper introduces the DAPE Dataset, a large-scale, well-curated, and richly annotated benchmark that addresses clear limitations of prior datasets and will be a valuable resource for the community.
2. The framework is shown to be a general-purpose enhancer that can be applied on top of various existing video editing models to improve their performance (as shown in Table 1).

**Weaknesses:**

1. The paper empirically demonstrates that "One-stage" joint training is suboptimal (Table 3) and attributes this to "negative interactions." However, it lacks a deeper analysis or intuition as to *why* this conflict occurs. Exploring the optimization dynamics (e.g., gradient conflicts) between the norm-tuning and the visual adapter would make the motivation even stronger.
2. While DAPE is a "parameter-efficient" method, the paper focuses primarily on performance (quality, consistency) gains. A more explicit quantification of the computational benefits (e.g., number of trainable parameters, training time per video) compared to full fine-tuning and the other PEFT baselines (like LoRA, Adapter from Table 4) would be beneficial.
3. The "Adjustable Norm-tuning" in Eq. 2 introduces a learnable $\gamma_0$ scaled by the original input $z_t$. While empirically effective, the paper could provide a clearer intuition for this specific formulation over other possible re-parameterizations.

**Questions:**

1. Could the authors please provide a quantitative comparison of the trainable parameter count and approximate training time for DAPE versus the baselines, particularly the other PEFT methods (LoRA, Adapter) and a full fine-tuning approach (if feasible)?
2. The paper provides a strong empirical case for the dual-stage design by showing "One-stage" training fails (Table 3). Could you offer more intuition on the nature of this "negative interaction"? Why do you believe the optimization for temporal consistency (norm-tuning) and visual detail (adapter) conflict when trained jointly?
3. The ablation for adapter position (Table 2) is excellent and shows that placing the adapter *only* in the first decoder block (position "5") is optimal. This is a very specific and localized insertion. Do you have a deeper architectural intuition for why this single, early-decoder block is so critical and effective, compared to a more distributed insertion (e.g., "5-7") or an encoder-side insertion?

---

### Official Review · Reviewer_WcBE · 2025-10-29

**Soundness:** 3
**Presentation:** 3
**Contribution:** 2
**Rating:** 4
**Confidence:** 5

**Summary:**

This paper introduces DAPE, a dual-stage, parameter-efficient fine-tuning (PEFT) framework for improving diffusion-based video editing. It aims to balance visual quality and computational efficiency by splitting the tuning process into two stages: first, an Adjustable Norm-tuning stage to enhance temporal consistency by fine-tuning normalization layer parameters; and second, a Visual Adapter-tuning stage to improve visual detail using a lightweight adapter module. The authors justify this sequential design by identifying a "negative interaction" that compromises performance when both components are trained simultaneously. The paper also contributes a new high-quality benchmark, the DAPE Dataset, to facilitate more robust evaluation.

**Strengths:**

- The DAPE framework is not a standalone model but a "plug-and-play" tuning method. The authors demonstrate this strength by applying DAPE to several existing baseline models (like Tune-A-Video, RAVE, and CCEdit) and showing consistent improvements.

- The quantitative results in Table 1 are convincing. They clearly show that applying DAPE ("DAPE (TAV)", "DAPE (RAVE)", etc.) almost universally improves the performance of the baseline models across a wide range of metrics, including temporal consistency (Warp Error, CLIP-Frame) and text-video alignment (CLIP-Text).

- The paper validates its method thoroughly using multiple quantitative metrics, qualitative still-frame comparisons (Figure 7), and a human user study (Figure 6) that shows a strong preference for DAPE's results in temporal consistency, alignment, and overall quality.

- The authors make two valuable contributions to the community by promising to release both their code for the DAPE framework and their newly curated DAPE Dataset.

**Weaknesses:**

- The primary weakness of the evaluation is the complete absence of video results. Video editing is an inherently dynamic task, and key artifacts like temporal flickering, incoherence, or unnatural motion can only be judged by viewing the output videos. The paper relies entirely on static frames (e.g., in Figure 7 and Figure 8) and aggregated user study scores. While the user study (Figure 11) asked participants to rate "overall smoothness (no distortion, flicker, etc.)", the reader cannot independently verify these claims. For a paper focused on temporal consistency, the lack of a supplementary webpage or embedded videos for qualitative assessment is a significant omission.

- The paper's core design—the dual-stage framework—is justified by a "negative interaction" observed when training the norm-tuning and adapter modules simultaneously (as shown in Table 3). However, the paper does not provide any technical analysis or insight into why this interaction occurs. The problem is stated, but not diagnosed. Understanding the root cause of this conflict (e.g., whether it's gradient-related, or if the modules compete for feature representation) is a non-trivial technical point that is left unaddressed.

**Questions:**

Please see weaknesses.

---

### Official Review · Reviewer_heX1 · 2025-10-31

**Soundness:** 2
**Presentation:** 1
**Contribution:** 2
**Rating:** 2
**Confidence:** 4

**Summary:**

To solve the issue of high computational costs and suboptimal performance in video editing task, this work designs a two-stage parameter-efficient fine-tuning framework. A high-quality dataset with rich annotations and editing prompts is also proposed. Experimental results prove the superiority of the proposed method.

**Strengths:**

- The writing is easy to understand, and the painting is well-drawn.
- The motivation of reducing computational costs and improve the performance for video editing task is effectiveness.

**Weaknesses:**

1.	The fourth case in Fig.1 change the appearance of the cat, however, the appearance of the edited squirrel is weird. The result also lacks dynamic motion of the changed subjects.
2.	In DAPE architecture, how does the model ensure both the quality of the temporal and visual feature learning if the framework uses a two-stage learning?
3.	In Adjustable Norm-tuning, the finetune of normalization parameters has been proved in previous methods, the proposed adjustable norm-tuning actually belongs to this paradigm.
4.	The proposed visual adapter adopts 5*5 conv, placed only in the first up-sampling cross-attention block. However, it acts as a bottleneck for edits that require global or high-frequency changes, like large object insertion, camera motion retargeting, and so on.
5.	In stage 1 and stage 2, there are no theoretical guarantee that the two objectives will not conflict when the data statistics change.
6.	Can we change the order of the two stages? Such as we train the visual adapter first, then freeze it and tune the norm layers?
7.	In experiments, except the listed baselines in Section 5.1, the latest models should be considered.
8.	Limited visualizations in Fig.7, additionally, as a video editing methods, it lacks videos files in the supplementary.
9.	Does the threshold hyperparameter in Eq. (v4) effect the training process? Related ablations should be considered.

**Questions:**

See weaknesses

---

### Official Review · Reviewer_ZJ8m · 2025-11-03

**Soundness:** 3
**Presentation:** 3
**Contribution:** 2
**Rating:** 4
**Confidence:** 4

**Summary:**

The paper proposes DAPE, a two-stage PEFT framework for video editing. The first stage introduces norm-tuning to enhance temporal consistency, while the second stage employs a vision-friendly adapter to improve visual quality. The authors also curate a new benchmark to address limitations in existing datasets, including limited category diversity, imbalanced object distributions, and inconsistent frame counts. Extensive experiments on existing datasets and the new benchmark demonstrate that DAPE improves temporal coherence and text-video alignment, outperforming prior state-of-the-art methods.

**Strengths:**

* Extensive experiments (baseline comparisons & ablation studies) support the claims.
* The paper is well-structured, easy to follow, and clearly written.
* The new benchmark enables more comprehensive evaluation of video editing methods.

**Weaknesses:**

* The framework (dual-stage tuning + adapters) appears as an incremental extension of existing diffusion adaptation techniques rather than a fundamentally new paradigm.
* The method assumes spatial and temporal features can be optimized independently, which may not hold for highly dynamic scenes. This may limit applicability to complex video content.
* Depth-wise 5×5 convolutions in visual adapters are empirically motivated, but there is no theoretical or architectural justification for the choice of kernel size or placement within the network.
* The paper uses an image-based diffusion model; it is unclear whether DAPE can be applied to DiT-based methods or video diffusion models (e.g., MotionDirector [1] and VMC [2]).
* Although framed as a parameter-efficient fine-tuning approach, the paper does not report compute or memory savings, leaving unclear whether the method is practically advantageous for high-resolution or long-duration videos.
* While the Huber loss addresses outliers, there is little discussion of performance on highly dynamic or out-of-distribution content.


[1] Zhao, Rui, et al. "Motiondirector: Motion customization of text-to-video diffusion models." ECCV 2024.
[2] Jeong, Hyeonho, et al. "Vmc: Video motion customization using temporal attention adaption for text-to-video diffusion models." CVPR 2024.

**Questions:**

* The reported improvements (vs. baselines) appear marginal; how sensitive are the evaluation metrics to these changes?
* Can DAPE be applied to video diffusion backbones beyond image-based models? and DiT backbones beyond UNet?
* Can the authors provide quantitative or qualitative evidence of compute and memory savings from the parameter-efficient fine-tuning approach? How does performance scale with longer video sequences, higher resolutions, or more complex motion?
* How robust is the method to domain shift or highly dynamic video content?

---

### Note · Authors · 2025-11-14

I have read and agree with the venue's withdrawal policy on behalf of myself and my co-authors.